# The localization of centromere protein A is conserved among tissues

Eleonora Cappelletti [1,6], Francesca M. Piras [1,6], Lorenzo Sola[1], Marco Santagostino [1], Jessica L. Petersen [2], Rebecca R. Bellone[3,4], Carrie J. Finno [3], Sichong Peng[3], Ted S. Kalbfleisch [5], Ernest Bailey[5], Solomon G. Nergadze[1] & Elena Giulotto [1✉]

Centromeres are epigenetically specified by the histone H3 variant CENP-A. Although mammalian centromeres are typically associated with satellite DNA, we previously demonstrated that the centromere of horse chromosome 11 (ECA11) is completely devoid of satellite DNA. We also showed that the localization of its CENP-A binding domain is not fixed but slides within an about 500 kb region in different individuals, giving rise to positional alleles. These epialleles are inherited as Mendelian traits but their position can move in one generation. It is still unknown whether centromere sliding occurs during meiosis or during development. Here, we first improve the sequence of the ECA11 centromeric region in the EquCab3.0 assembly. Then, to test whether centromere sliding may occur during development, we map the CENP-A binding domains of ECA11 using ChIP-seq in five tissues of different embryonic origin from the four horses of the equine FAANG (Functional Annotation of ANimal Genomes) consortium. Our results demonstrate that the centromere is localized in the same region in all tissues, suggesting that the position of the centromeric domain is maintained during development.

[1] Department of Biology and Biotechnology, University of Pavia, Pavia, Italy. [2] Department of Animal Science, University of Nebraska–Lincoln, Lincoln, NE, USA. [3] Department of Population Health and Reproduction, School of Veterinary Medicine, University of California-Davis, Davis, CA, USA. [4] Veterinary Genetics Laboratory, School of Veterinary Medicine, University of California-Davis, Davis, CA, USA. [5] Gluck Equine Research Center, University of Kentucky, Lexington, KY, USA. [6] These authors contributed equally: Eleonora Cappelletti, Francesca M. Piras. ✉email: elena.giulotto@unipv.it

The centromere is a specialized nucleoprotein structure of the eukaryotic chromosome. It is the site of kinetochore assembly required for proper chromosomal segregation during cell division. Despite the fact that centromeric function is well conserved along the evolutionary scale, centromeric DNA sequences are highly divergent among taxa and are not necessary nor sufficient to specify the centromeric function[1–3]. This paradox is explained by the well-established knowledge that the centromeric function is epigenetically specified and, thus, not determined by the underlying DNA sequence[4]. Indeed, CENP-A, the centromere-specific variant of the histone H3, is the epigenetic mark of functional centromeres[5].

In mammals, centromeric DNA typically consists of large arrays of tandemly repeated sequences (satellite DNA), which are extremely divergent and represent the most rapidly evolving DNA sequences in eukaryotic genomes[6]. The presence of such sequences has so far posed a barrier for carrying out a comprehensive molecular analysis of these enigmatic loci.

Although satellite DNA is typically present in mammalian centromeres, we previously demonstrated that in the genus *Equus* (horses, asses and zebras), several centromeres are completely satellite-free, thus representing a unique model for dissecting the molecular architecture of mammalian centromeres[7–18].

The horse reference genome was the first Perissodactyl genome assembly to be published and is the most curated reference genome among equids[11,19,20]. The release of the horse reference genome was accompanied by the discovery, carried out by our group, of a satellite-free centromere, identified on chromosome 11 (ECA11). Satellite-free neocentromeres have been previously described in sporadic human clinical samples[2,3,21,22] while the ECA11 centromere was the first centromere devoid of satellite DNA to be found stably present in a vertebrate species, demonstrating that a natural centromere can exist without satellite DNA[11]. This centromere emerged recently during evolution as a result of centromere repositioning, which is the shift of the centromere position along the chromosome without sequence rearrangements. This event occurred many times during the rapid karyotypic evolution of equid species and, together with Robertsonian fusion, led to the formation of a large number of satellite-free centromeres in the genus *Equus*[7,11,13,15,23].

We subsequently showed that the position of the CENP-A binding domain of ECA11 is not fixed but slides within an about 500-kb region and different positional alleles or epialleles were identified in different individuals[7–9]. This phenomenon termed centromere sliding was also described in other equid species[7,13]. We also demonstrated that these epialleles are inherited as Mendelian traits, but their position can slide in one generation[7]. On the contrary, centromere position is stable during mitotic propagation of cultured cells, suggesting that sliding may presumably take place either during meiosis or in early embryogenesis[7].

To our knowledge, no prior studies have compared the position of centromeric domains in different tissues. The results of such studies would shed light on the mechanisms of centromere propagation during development.

An answer to this open question may come from the Functional Annotation of ANimal Genomes (FAANG) project[10,24,25]. This international collaboration was established in 2015 and aims to systematically annotate animal genomes. As part of the international initiative, the equine FAANG group has led annotation efforts of the horse genome[10,26–31]. The first stage of the equine FAANG project was to generate biobanks of tissues and cell lines from four comprehensively phenotyped adult animals (ECA_UCD_AH1- ECA_UCD_AH4)[32,33]. These healthy animals —two mares and two stallions—were selected from the same breed (Thoroughbred) as Twilight, the mare used to obtain the horse reference genome[11,19,20].

Here, we mapped the position of the ECA11 CENP-A binding domain in a fibroblast cell line from Twilight, improving the reference sequence of the centromeric region of chromosome 11 in the EquCab3.0 horse reference assembly. We then mapped the position of the ECA11 centromeric domain in different tissues and cell lines of the four FAANG horses.

## Results

**Improvement of the reference sequence in the centromeric region of chromosome 11.** The horse reference genome was obtained from a Thoroughbred mare (Twilight) and is the most curated genome sequence among equids[11,19]. We previously demonstrated that the centromere of horse chromosome 11 is devoid of satellite DNA[11]. We also demonstrated that the CENP-A binding domain at this centromere is not fixed in the horse population but slides within a 500-kb genomic region[9]. To map the precise position of the CENP-A binding domain on chromosome 11 in Twilight, we performed a ChIP-seq experiment with an anti-CENP-A antibody on a primary skin fibroblast cell line from this individual. ChIP-seq reads were mapped to the last release of the horse reference genome (EquCab3.0) and, as expected, an enrichment peak was obtained on chromosome 11 (Fig. 1a). The CENP-A binding domain was localized in the genomic window in which centromeric domains were known to slide in the horse population[9]. However, the CENP-A enrichment peak was irregular and contained coverage dips (Fig. 1a, top panel), suggesting that the sequence underlying the centromeric domain is misassembled in EquCab3.0.

With the goal of determining more precisely the sequence of the centromeric region, we used our paired-end ChIP and input reads and the publicly available PacBio (SRR6374292) and Illumina WGS (SRR6374293) reads to assemble the 617-kb genomic segment containing the CENP-A binding domain of Twilight (NCBI Accession number OQ679756) using an iterative chromosome walking approach, as previously described[7,13]. We then corrected the EquCab3.0 reference by removing the centromeric locus (chr11:27592872-28352430) and replacing it with the newly assembled sequence.

In the original EquCab3.0 assembly, the region chr11:27707536-27808813 shared high sequence identity with the region chr11:27809814-27911066 and with the entire sequence of the unplaced contig NW_019645621.1. The PacBio long reads and Illumina WGS reads allowed us to demonstrate that at this locus no sequence duplication is present. In addition, two sequence gaps (chr11:27808813-27809813 and chr11:28295240-28296240) were present and the first one falls within a coverage dip of the CENP-A peak. In the resulting reference genome that we called EquCab3.0_cen, we corrected these misassemblies and curated sequence gaps. The length of the locus (from 759559 to 617491 nt) as well as the number of unknown (N) nucleotides per 100 kb (from 263.31 to 33.52) decreased. We then remapped our ChIP-seq reads on EquCab3.0_cen. The peak profile visualized on the new reference genome (Fig. 1a, bottom panel) was greatly improved as well as the mapping qualities of reads (Supplementary Fig. 1). Two relatively well-defined peaks were identified, suggesting the presence of different epialleles on the two homologs.

To test whether EquCab3.0_cen is a valid reference for the centromeric region of horse chromosome 11, we compared the peaks of ChIP-seq reads mapped on the EquCab3.0 and on the EquCab3.0_cen reference genomes from six additional horses. As shown in Fig. 1b, the shapes of the enrichment peaks on EquCab3.0_cen are more regular than those obtained using the EquCab3.0 reference. In particular, Horse S, Horse C and Horse D display interrupted peaks on the EquCab3.0 reference. Using the

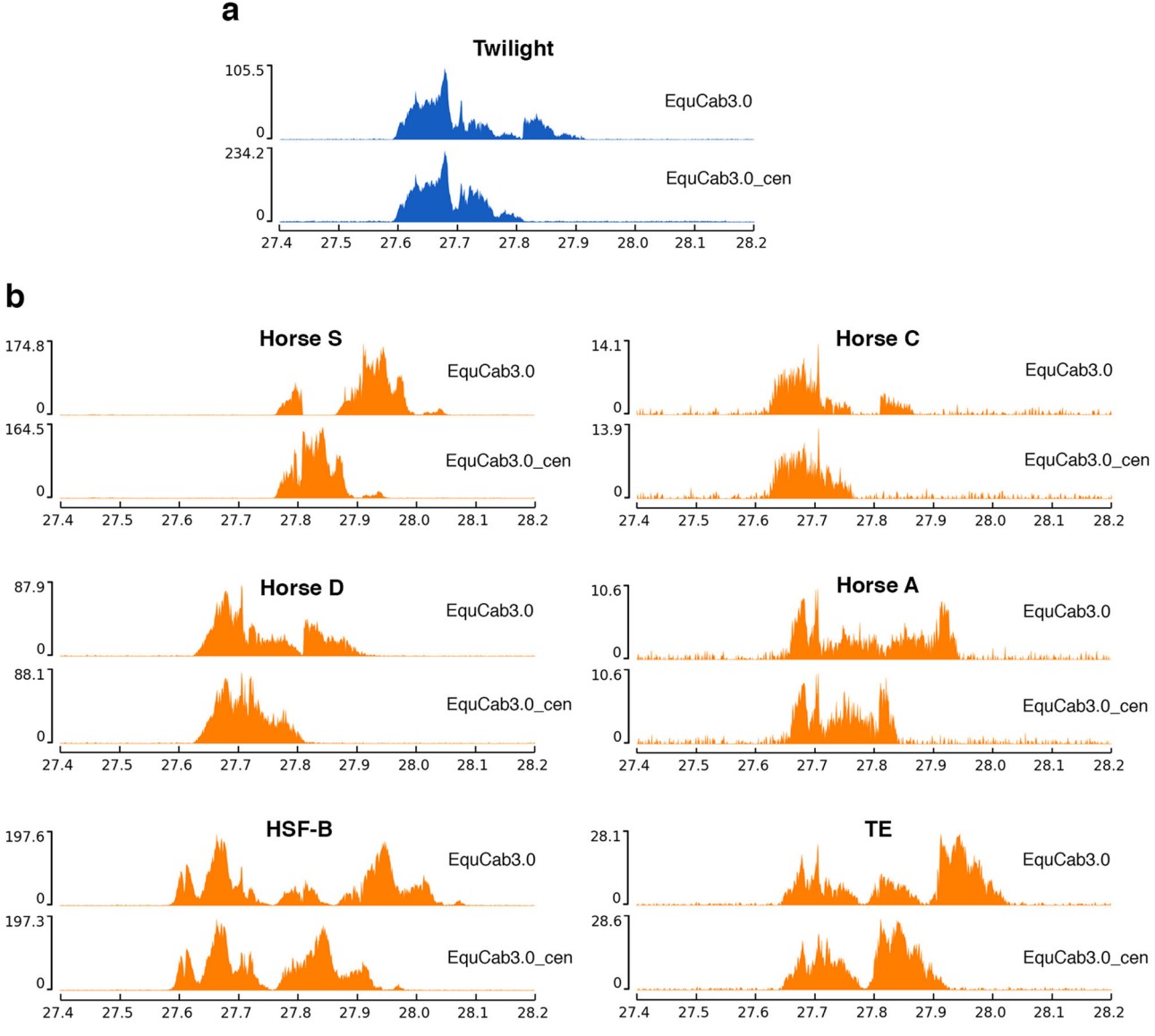

**Fig. 1 Improvement of the reference sequence in the ECA11 centromeric region. a** ChIP-seq reads from primary fibroblasts of Twilight were mapped on the EquCab3.0 (upper panel) or on the EquCab3.0_cen (lower panel) references. The CENP-A enriched domain is visualized as a peak. The y-axis reports the normalized read counts whereas the x-axis reports the coordinates on the reference genome. **b** ChIP-seq reads from the primary fibroblasts of six different horse individuals were mapped on the EquCab3.0 (upper panels) or on the EquCab3.0_cen (lower panels) references. The CENP-A enriched domains are visualized as a peak. The y-axis reports the normalized read counts whereas the x-axis reports the coordinates on the reference genome.

EquCab3.0_cen reference, several coverage dips were curated and the peaks become single Gaussian-like peaks covering about 200 kb on the EquCab3.0_cen reference. Horse A shows a broad irregular peak on the EquCab3.0 reference while using the EquCab3.0_cen reference, a more regular and compact peak, occupying about 200 kb, was obtained. After mapping the reads on EquCab3.0, horses HSF-B and TE display three peaks of different heights while, after mapping on EquCab3.0_cen, two peaks were observed, suggesting that different epialleles for CENP-A binding are present on the two homologous chromosomes.

**CENP-A binding domains in different tissues of the four FAANG horses**. As part of the FAANG initiative, the equine FAANG community aims to functionally annotate the horse genome[10]. The first stage of the equine FAANG initiative was to generate a biobank of reference tissues and cell lines from four comprehensively phenotyped animals: two Thoroughbred mares ECA_UCD_AH1 (AH1) and ECA_UCD_AH2 (AH2)[33] and two

Thoroughbred stallions ECA_UCD_AH3 (AH3) and ECA_UC-D_AH4 (AH4)[32].

In this study, we characterized the position of the CENP-A binding domain of the ECA11 satellite-free centromere in the four FAANG individuals by performing ChIP-seq experiments with an anti-CENP-A antibody on fibroblast cell lines. ChIP-seq reads were mapped on the EquCab3.0_cen reference (Fig. 2a). As expected, the CENP-A binding domains were localized in the genomic window in which we previously showed that the centromeric domains could slide in the horse population[9]. Interindividual variation for the position of CENP-A binding on chromosome 11 was observed among the four horses (Fig. 2a). Horse AH1 shows a main enrichment peak of about 140 kb and a secondary peak of about 40 kb. The two peaks are separated by a 50 kb region that is not bound by CENP-A in this horse, while it is included within the CENP-A binding domains in the AH2 and AH3 horses. A single rather regular peak covering about 150 kb is observed in the AH2, AH3 and AH4 horses.

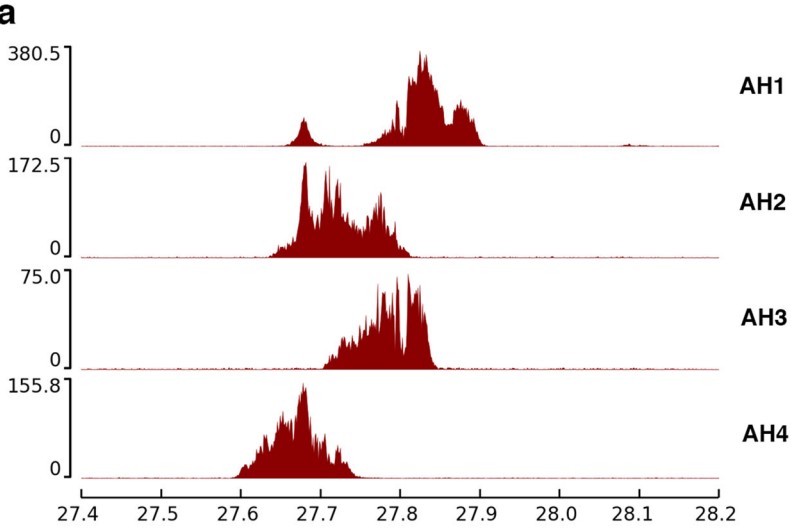

We previously demonstrated that the position of the satellite-free centromere of chromosome 11 slides in the horse population and that the epialleles for CENP-A binding are inherited as Mendelian traits, but their position could slide in one generation[7,9]. Conversely, the position of the centromere is stable during mitotic propagation of cultured cells, suggesting that sliding may presumably take place during meiosis or early embryogenesis[7].

To test whether the centromere position is conserved during development or whether it can slide during tissue differentiation, we performed ChIP-seq experiments with the anti-CENP-A antibody on four tissues of different embryonic origin (ovary/testis, liver, lamina and brain) from the four FAANG individuals. ChIP-seq reads were mapped on the EquCab3.0_cen reference genome. We then compared the position of CENP-A binding

**Fig. 2 ChIP-seq characterization of the ECA11 satellite-free centromere in the FAANG horses. a** ChIP-seq characterization of the ECA11 satellite-free centromere in the fibroblast cells lines from the FAANG mares (AH1 and AH2) and the FAANG stallions (AH3 and AH4). ChIP-seq reads from primary fibroblasts were mapped on the EquCab3.0_cen reference. The CENP-A enriched domains are visualized as peaks. The y-axis reports the normalized read counts whereas the x-axis reports the coordinates on the reference genome. **b** ChIP-seq profiles of the CENP-A binding domain on ECA11 in the fibroblast cell line (top) and in four tissues of different embryonic origin from the FAANG mares (AH1 and AH2) and FAANG stallions (AH3 and AH4). Color code refers to the embryonic origin. The y-axis reports the normalized read counts whereas the x-axis reports the coordinates on the EquCab3.0_cen reference genome. The scale of the y-axis is not the same across samples to highlight the position of the peak rather than its height. **c** Enrichment peaks obtained using SICER2.

domains in the different tissues and in the fibroblast cell line of the same individual. As shown in Fig. 2b and Supplementary Fig. 2, the position of the CENP-A binding domain in the four tissues and in the fibroblast cell line of each individual is conserved as confirmed by the results of peak calling (Fig. 2c). This result suggests that the position of the centromeric domains was maintained during development.

## Discussion

In 2009, the genome sequence of the Thoroughbred mare Twilight was published and established a reference for the domestic horse, EquCab2.0[11,20]. The genome of Twilight was recently re-sequenced and assembled, resulting in EquCab3.0 assembly, a reference genome improved in terms of contiguity and composition[19,20]. A unique feature of the horse genome that we discovered in the first assembly[11] was the presence of a centromere completely devoid of satellite repeats at chromosome 11. This centromere was the first example of a natural satellite-free centromere described in a vertebrate species[11]. We then demonstrated that the position of the CENP-A binding domain of ECA11 is not fixed but can slide within an about 500-kb region in different individuals, giving rise to positional alleles or epialleles[7,9].

In this study, we improved the reference sequence of chromosome 11 in the EquCab3.0 genome assembly by replacing the region corresponding to the centromeric domain of Twilight with the sequence that we assembled from our ChIP-seq reads and publicly available short-read and long-read sequencing data. This new version of EquCab3.0, denoted as EquCab3.0_cen, can be considered a refined assembly regarding the centromeric region of chromosome 11. Using this new reference sequence, the mapping of our ChIP-seq reads in this centromeric region improved, allowing us to better define the position of CENP-A binding domains at the centromere of ECA11 in Twilight and in the six additional individuals (Fig. 1). The shape and the extension of the CENP-A binding domain suggest that Twilight carries different epialleles on the two ECA11 homologs that are partially overlapping on the reference genome. We previously demonstrated that CENP-A binding domains are inherited as Mendelian traits and, indeed, the peak of horse AH4, which is the son of Twilight[32], is overlapping with its mother peak. It will be interesting to extend this analysis to horses from different breeds in order to identify more epialleles, to test whether some breeds are characterized by the presence of specific epialleles and to determine the degree of variation in the population of this particular polymorphism.

We then characterized the ECA11 CENP-A binding domain in the four FAANG horses. In Fig. 2a, the position of the enrichment peaks in the fibroblasts of the four horses are compared in the EquCab3.0_cen reference genome indicating that each individual is characterized by a different profile. Three out of the four FAANG horses display single peaks of about 150 kb. Since we previously showed that each epiallele covers a region of about 100 kb, it is likely that, in these horses, the two homologs carry their CENP-A binding domain in partially overlapping regions.

Differently, the peak profile of the ECA_UCD_AH1 mare is peculiar with a main enrichment peak of about 140 kb which is 50 kb away from a secondary peak of about 40 kb. The different length and enrichment of the two peaks makes it unlikely that they correspond to different epialleles on the two homologous chromosomes. No other horse with a CENP-A binding domain in this region shows this distinctive profile, leading us to hypothesize that this horse carries a chromosomal rearrangement involving the 50 kb region where ChIP-seq reads were not enriched. Similarly, we cannot exclude that some of the peak irregularities of the other horses may be due to specific sequence rearrangements compared to the reference genome.

It was proposed that the emergence of new centromeres during evolution may be triggered by DNA fragility. Since sites of breakage are recurrent during evolution and some of them tend to be used for centromere formation[34,35], it is possible that, also at ECA11, DNA breaks and rearrangements may have occurred. If this was the case, we may expect some variability, due to sequence rearrangement of this region, in the horse population, which may bias alignments of CENP-A ChIP-seq data from other horses to the Twilight-derived EquCab3.0 or EquCab3.0_cen assemblies.

The observation of multiple epialleles in the satellite-free centromeric regions of horse chromosome 11 led us to investigate when the shift of CENP-A domains can occur. We previously demonstrated that, while the centromere position was stable during cell propagation in culture, CENP-A binding domains were inherited as Mendelian traits but could slide in one generation[7]. In addition, we observed instances of substantial centromere movement, on the order of 50–80 kb, that occurred in a single generation. This is the type of shift that we were expecting to observe if centromere sliding occurred during development. The extent of this movement was never extreme with the centromeric domain of the offspring at least partially overlapping the domain of the parent. This finding suggested that, in a single generation, a fraction of CENP-A nucleosomes slightly move from the original position and that, in the course of several generations, these movements may accumulate, giving rise to nonoverlapping epialleles. This phenomenon may take place during germline differentiation, meiotic division, fertilization or early developmental stages[7]. These stages are indeed characterized by extensive chromatin remodeling and epigenetic reprogramming which may be accompanied by CENP-A mobilization[36,37].

Mechanisms of CENP-A chromatin deposition and propagation are well characterized in somatic cells during mitosis[38,39] but remain controversial in the germline[40,41] although it is well described that CENP-A nucleosomes are maintained through the widespread histone-to-protamine replacement in spermatogenesis[40,42]. Similarly, it was shown that CENP-A nucleosomes are retained at centromeres during the prolonged prophase I arrest[40,41,43]. It has been proposed that the structural rigidity of CENP-A nucleosomes is the key for explaining centromere inheritance during mammalian gametogenesis[40,41]. However, the fidelity of CENP-A deposition is poorly understood during the different stages of meiosis[44] and we cannot exclude

that centromere sliding may occur during the meiotic division itself. Notably, we previously detected centromere sliding from parent to offspring in both the maternal and the paternal line[7].

Alternatively, centromere sliding may occur during the early embryo development and tissue differentiation, which are characterized by massive chromatin remodeling and active DNA demethylation and remethylation[45–47]. In this work, thanks to the availability of the repository of horse tissues and cell lines collected from four Thoroughbred horses in the FAANG consortium[32,33], for each individual, we compared the position of the enrichment peaks in different tissues. We demonstrated that the position of CENP-A binding at chromosome 11 is conserved among all tissues and cell lines from ectodermal (lamina and brain), mesodermal (fibroblasts, ovary and testis) and endodermal (liver) embryonic origins. It is worth noticing that the CENP-A peak from the testis of stallion AH3 shows a tail that is not detected in the other tissues. We cannot definitely conclude whether this tail is part of the background or is due to the presence of a sub-population of cells in which sliding may have occurred. It is tempting to speculate that the shape of the CENP-A binding domain from the testis of AH3 may be due to the presence of a fraction of mature spermatozoa and cells at different stages of meiosis in which sliding may have occurred. An interesting development of the work presented here will be to compare CENP-A binding profiles from samples of spermatozoa with other tissues. However, a clear answer on the possibility of sliding in meiosis could come from ChIP-seq on single spermatozoa and oocytes, which will be the aim of future work. While the number of horses and tissues is relatively limited, the fact that all four samples showed consistent results across the different tissues studied supports that the centromere position is conserved during development and that the phenomenon of centromere sliding does not occur during tissue differentiation. Therefore, centromere sliding is presumably occurring during the unique epigenetic transactions of meiosis.

## Methods

**Cell lines**. The primary fibroblast cell line from Twilight was kindly provided by Donald Miller (Cornell University, Ithaca, NY)[19]. Primary fibroblasts from ECA_UCD_AH1, ECA_UCD_AH2, ECA_UCD_AH3 and ECA_UCD_AH4 were established from skin biopsies within the FAANG consortium[32,33].

The cells were cultured in high-glucose DMEM medium supplemented with 20% fetal bovine serum, 2 mM L-glutamine, 1% penicillin/streptomycin and 2% non-essential amino acids. Cells were maintained at 37 °C in a humidified atmosphere of 5% $CO_2$.

**ChIP-seq with anti-CENP-A antibody and downstream bioinformatic analysis**. Chromatin from about 50 million primary fibroblasts was cross-linked with 1% formaldehyde, extracted, and sonicated to obtain DNA fragments ranging from 200 to 800 bp. Chromatin from brain, lamina, liver, ovary and testis was extracted and sheared by Diagenode ChIP-Seq Profiling Service (Diagenode, Cat# G02010000, Liège, Belgium) as previously described[26,28]. For ChIP experiments, we used variable amounts of chromatin ranging from 4 to 13 µg. Complete summary of the final protocols used for all tissues can be accessed at ftp://ftp.faang.ebi.ac.uk/ftp/protocols/assays/. The tissue samples were obtained from the FAANG equine biobank[32,33]. The establishment of this biobank was reviewed and approved by UC Davis Institutional Animal Care and Use Committee.

Immunoprecipitation was performed as previously described[7,48] by using an anti-CENP-A serum[8]. In particular, the immunocomplex was purified using A/G beads (nProtein A Sepharose™ 4 Fast Flow/Protein G Sepharose™ 4 Fast Flow, GE Healthcare). After reverse cross-linking, carried out overnight at 65 °C, immunoprecipitated and input DNAs were extracted with the Wizard Genomic DNA Purification Kit (Promega) according to the manufacturer's instructions. Paired-end sequencing was performed with Illumina HiSeq2500 and Illumina NovaSeq6000 platforms by IGA Technology Services (Udine, Italy). ChIP-seq reads from horse S (accession numbers SRX2789367 and SRX2789358), A (accession numbers SRX2789324 and SRX2789325), C (accession numbers SRX2789347 and SRX2789336), D (accession numbers SRX2789370 and SRX2789369) and TE (accession numbers SRX6609390- SRX6609393) were previously mapped on EquCab2.0[7,8]. The ChIP-seq reads from horse HSF-B were obtained from a previously described fibroblast cell line[9] and are available at NCBI SRA archive (accession numbers SRR23995266 and SRR23995265).

Reads were aligned with paired-end mode to the EquCab3.0 or EquCab3.0_cen reference genomes with Bowtie2 aligner using default parameters (2.4.2 version)[49,50]. Normalization of read coverage of the ChIP datasets against the input datasets was performed using bamCompare available in the deepTools suite (3.5.0 version)[51] using RPKM (Reads Per Kilobase per Million mapped reads) normalization in subtractive mode. Peaks were obtained with pyGenomeTracks (3.6 version)[52,53]. Peak calling was performed using SICER2[54] using -w 200 and -g 1000 parameters and filtering for islands with false discovery rate (FDR) less than or equal to 0.01.

**Assembly of ECA11 centromeric region and improvement of the EquCab3.0 reference genome**. The assembly of the CENP-A binding domain from Twilight was performed using an iterative chromosome walking approach based on the paired-end ChIP-seq reads, that we previously applied to the assembly of donkey, Burchell's zebra and Grevy's zebra centromeric regions[7,13]. We used our ChIP and Input reads together with the publicly available Illumina WGS (SRR6374293) and PacBio (SRR6374292) reads from this individual. Paired-end Illumina WGS reads were trimmed using Trim Galore! (Galaxy Version 0.6.7+galaxy0) and aligned with Bowtie2 aligner using default parameters (2.4.2 version). PacBio reads were aligned using minimap2 (Galaxy Version 2.26+galaxy0)[55,56] using map-pb profile. Aligned reads were visualized using Integrative Genome Viewer (IGV, Version 2.9.2) on the EquCab3.0 reference genomes. We retrieved the consensus of the mapped reads using the Copy consensus sequence function of IGV. Consensus bases other than ACGTN were corrected by a visual inspection of reads aligned to the reference genome. We then proceeded to refine these draft sequences, resolving misassembled regions by de novo assembly of raw paired-end reads using a walking approach. To this end, we joined read pairs of ChIP and Input datasets using FASTQ joiner (Galaxy Version 2.0.1.1 + galaxy0)[57]. Queries of 60–95 bp, flanking gaps or misassembled regions of the draft sequences, were searched in the joined paired-end reads using the Grep command. Retrieved reads were aligned using MultAlin[58] and a new query was designed on the newly obtained sequence. This procedure was reiterated to resolve gaps and misassembled regions that were present in the draft consensus. Mapping qualities of reads aligned in the centromeric region that we assembled (EquCab3.0_cen) compared to the original EquCab3.0 reference were extracted using Samtools (version 1.15.1)[59] and plotted using the R software package ggplot2. To refine the reference sequence of chromosome 11 in the EquCab3.0 assembly, we first run BLAT (v. 36)[60] using the assembled contig as a query to identify its misassembled counterpart in the EquCab3.0 reference. The EquCab3.0 sequence was then removed and substituted with

the newly assembled centromeric contig using SAMtools (version 1.15.1)[59] and Bash commands. The number of N's per 100 kb in the original sequences and in the newly assembled contig was obtained using Quast Genome assembly Quality (Galaxy Version 5.0.2+galaxy4 or online version available at http://cab.cc.spbu.ru/quast/).

**Statistics and reproducibility.** The identification of domains enriched for CENP-A was performed using SICER2[54]. Significant islands were filtered by FDR. Only regions with FDR less than or equal to 0.01 were considered.

**Reporting summary.** Further information on research design is available in the Nature Portfolio Reporting Summary linked to this article.

## Data availability

Raw sequencing data from this study are available in the NCBI BioProject database (https://www.ncbi.nlm.nih.gov/bioproject/) under accession numbers PRJNA945609 and PRJNA949688. The assembled centromeric regions of chromosome 11 from this study are available in the NCBI Nucleotide database (https://www.ncbi.nlm.nih.gov/nucleotide/) under accession number OQ679756. In this work we also used publicly available WGS (SRR6374293), ChIP-seq (SRX2789367, SRX2789358, SRX2789324, SRX2789325, SRX2789347, SRX2789336, SRX2789370, SRX2789369, SRX6609390, SRX6609391, SRX6609392 and SRX6609393) and PacBio reads (SRR6374292).

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

## Acknowledgements

We would like to thank Donald Miller (Cornell University, USA) for providing us with the fibroblast cell line from Twilight and Francesco Lescai (University of Pavia) for helpful suggestions during the revision of the manuscript. This research was funded by Animal Breeding and Functional Annotation of Genomes (A1201) Grant 2019-67015-29340/Project Accession 1018854 from the USDA National Institute of Food and Agriculture and by the Italian Ministry of Education, University and Research (MIUR) (Dipartimenti di Eccellenza Program (2018–2022)—Department of Biology and Biotechnology "L. Spallanzani," University of Pavia). The Galaxy server that was used for some calculations is in part funded by Collaborative Research Centre 992 Medical Epigenetics (DFG grant SFB 992/1 2012) and German Federal Ministry of Education and Research (BMBF grants 031 A538A/A538C RBC, 031L0101B/031L0101C de.NBI-epi, 031L0106 de.STAIR (de.NBI)).

## Author contributions

E.G. conceived the study and supervised all experiments. E.C., F.M.P. and S.G.N. carried out experiments and bioinformatic analyses and contributed to result interpretation and figure preparation. L.S., M.S. and S.P. contributed to bioinformatic analysis. E.G., E.C., F.M.P., S.G.N., L.S. and M.S. participated in discussions and result interpretation. E.G., E.C. and F.M.P. wrote the manuscript. J.L.P., R.R.B. C.J.F, T.S.K., E.B. and E.G. provided biological material and funding.

## Competing interests

The authors declare no competing interests.
