## [Peer Review File · Communications Biology]

Reviewers' comments:

Reviewer #1 (Remarks to the Author):

THE LOCALIZATION OF CENTROMERE PROTEIN A IS CONSERVED AMONG TISSUES

Cappelletti et al.

Centromere identity is defined epigenetically by the presence of CENP-A, but the highly repetitive satellite DNA occurring at most vertebrate centromeres has--until recently--precluded determination of precise CENP-A position within centromeric chromatin. Circumventing that issue, the authors utilize a previously characterized evolutionarily new horse centromere that lacks alpha satellite or other repetitive DNA to precisely examine CENP-A position in related horses and between tissues of the same horse. Their findings highlight the sliding of CENP-A position at non-repetitive centromere ECA11 between individuals of the same species and the maintenance of those epialleles within tissues of different embryonic origin in the same individual. These findings suggest that individual CENP-A epialleles are determined prior to tissue specification during horse development. While this finding is novel and interesting, the authors do not provide evidence as to when the slide does happen if it doesn't happen during development or in mitotic divisions, but does appear to happen within one generation. The authors suggest that the centromere slide may happen during meiosis, but this suggestion is not followed through. Mechanisms that drive such a slide are not explored or discussed. The genomic re-assembly method of the ECA11 centromere region is concerning, as explained below and can affect the results presented.

Major Points:

1. Given the documented existence of discrete CENP-A binding regions and gaps within the CENP-A binding region at non-repetitive centromeres in humans and horses (Hasson et al., *Nat Struct. Mol. Biol.* 2013; Nergadze et al., *Genome Res.* 2018), it is unclear why the authors believed that gaps within CENP-A binding domains found at ECA11 by aligning CENP-A ChIP-seq to EquCab3.0 indicated an issue with the integrity of the EquCab3.0 reference. It's possible that these discrete CENP-A positions represent maternal and paternal CENP-A epialleles at ECA11. For instance, even after reassembly of the ECA11 locus, horses HSF-B and TE still have two distinct CENP-A loci.

2. The authors methodology for improving the sequence of ECA11 in EquCab3.0 is an "iterative chromosome walking approach" that utilizes their CENP-A ChIP-seq reads. However, re-building this locus using only reads that bind CENP-A selectively removes any regions/reads that do not bind CENP-A, which could merge discrete CENP-A regions and bias the representation of CENP-A position, and is not necessarily the true genomic assembly of that region. The authors do not provide enough information on how this was performed and therefore it is hard to understand if this is indeed an improvement of the horse ECA11 sequence assembly. It would be helpful to provide more details on this analysis, including a figure depicting what parts of the original assembly are lost in the new EquCab3.0_cen assembly and how CENP-A read positions change between the two assemblies. Specifically, it appears the region between ~27.8 and ~27.9 in the EquCab3.0 assembly are lost/rearranged in the EquCab3.0_cen assembly. Are CENP-A reads mapping here lost or relocated to new positions? How does this affect position of CENP-A reads that aligned to this location in the Horse S, D, C, A, HSF-B and TE samples?

Given the plethora of newer sequencing methodologies currently available, a better approach would have been to simply perform a long-read sequencing approach, such as Oxford nanopore sequencing or Pac Bio long-read sequencing to re-assemble that region.

3. The authors chose to perform cross-linked ChIP-seq for CENP-A instead of native ChIP-sequencing. Since CENP-A is a histone, native ChIP-seq can be done for CENP-A and is the preferred choice that allows titrated MNase digestion, obtaining mono-nucleosome pool (instead of longer stretches of chromatin) before IP, and therefore higher resolution for mapping of CENP-A binding. By choosing fixed chromatin, shearing and mapping of longer chromatin fragments the authors reduce the resolution of CENP-A mapping which is important for their study.

4. The author suggests that the discrete regions of CENP-A binding in some of the horses (Fig. 1B, HSF-B and TE horses) can indicate maternal and paternal alleles. If they have analyzed and reassembled that region of the horse genome using long-read sequencing approaches, that could have led potentially to the identification of possible SNPs in that region, that could have been used to phase the CENP-A ChIP-seq reads and obtain separate maternal or paternal CENP-A binding patterns.

5. The emergence of evolutionarily new centromeres like ECA11 is thought to occur at regions where DNA is fragile or frequently transcribed, allowing opportunities for incorporation of CENP-A. Though ECA11 is not a repetitive centromere, other evolutionarily young horse centromeres that do not occur at satellite DNA have acquired repetitive elements and previous work suggests that ECA11 is likely to also gain repetitive DNA elements as it continues to mature. This, and the authors finding that the AH1 mare may carry a chromosomal rearrangement at the ECA11 locus suggests that this region may also be fragile or prone to DNA rearrangements. Given this knowledge, the authors should discuss the potential differences in the underlying DNA sequences between different horses at the ECA11 locus and how this may bias alignments of CENP-A ChIP-seq data from other horses to the Twilight-derived EcuCab3.0 or EcuCab3.0_cen assemblies.

6. In figure 2, there seems to be some small-scale differences or sliding of CENP-A position between fibroblasts and testis samples of stallions AH3 and AH4. It would be helpful if the authors provided a track of significantly enriched CENP-A peaks relative to background to help clarify this. Differences in CENP-A position between fibroblasts and testis (does this sample contain spermatozoa?) might be important for helping to define when CENP-A position changes between horse generations.

7. The authors found that there is no difference in CENP-A binding in different tissues of the same individual suggesting that CENP-A binding pattern is determined prior to tissue differentiation. This is an interesting and novel finding. However, the authors showed that centromere sliding can happen in one generation (compare CENP-A binding pattern of Twilight's fibroblasts in Fig. 1a with the pattern of her son AH4 in Fig. 2a), and previously showed that the centromere position is stable during mitotic divisions, suggesting that the sliding may occur during meiosis. The authors did not explore further this option.

Minor Points:

1. The authors state in lines 57-59 that ECA11 is the first centromere devoid of alpha satellite DNA to be detected in vertebrates. However, many human centromeres devoid of alpha satellite DNA have been documented prior to this finding (Amor and Choo, *Am J Hum Genet.* 2002), with the first being in 1993 (Voullaire et al., *Am J Hum Genet.*).

2. It would have been nice to show the CENP-A binding pattern of Twilight's fibroblasts (Fig. 1a) directly above the pattern of her son (Fig. 2a,b) to better present the centromere sliding within one generation.

Reviewer #2 (Remarks to the Author):

This manuscript investigates centromere positioning and inheritance in horses. Uniquely, horses harbour a satellite free centromere on chromosome 11, mapped by the binding of the centromeric histone CENP-A. The discovery of this natural satellite free centromere has enabled investigations into centromere specification and transgenerational inheritance at the genomic level. It was previously shown that the domain of CENP-A binding is inherited, but that it can slide within a 500 kb window in a single generation. As centromere sliding is not observed in mitotic cells in culture, the authors propose that it occurs in meiosis or embryogenesis. To address this possibility, the authors investigate whether CENP-A position changes in differentiated tissues of the same animal. Overall, the data

supports the conclusion that CENP-A position does not change, however a more detailed description of CENP-A profile in the investigated tissues would be informative here (see point 2 below).

This study is well written and presented and the experiments are carried out at a high standard. It presents two major findings:

First, presented in Figure 1, the authors use newly generated CENP-A paired end ChIP seq data to redefine the centromeric region of chromosome 11 in the latest release of the Twilight horse reference genome (EquCab3.0). Based on maps generated for Twilight and 6 additional horses, this new and corrected centromere 11 reference sequence, called EquCab3.0_cen, contains more regular CENP-A peaks that are less interrupted and more Gaussian in shape. Going forward, this is an important resource for the community.

Second, to test whether centromere position can slide during tissue differentiation, the authors map and compare CENP-A position by ChIP-seq in five tissues from four individual horses (extensively phenotyped as part of the FAANG project). The tissue selected were from different embryonic origin – endodermal (liver), mesodermal (ovary/testes) and ectodermal (lamina and brain). Results presented in Figure 2b showed that, in general, the position of the centromere did not move compared to the control fibroblast cell line. However, there does appear to be more 'background signal' in all data from tissues, particularly in the case of the brain and ovaries/tissues. Can the authors comment on this? Also in the ovary and tissue samples, is it possible that these tissues contain meiotic cells? This would be important to know, especially as the CENP-A signal appears to occupy a larger domain with less sharp peaks e.g. sample AH4.

We would like to thank the Reviewers for their useful and stimulating suggestions that allowed us to improve the manuscript.

In the new version of the manuscript we modified the text according to the comments of the Reviewers and added two Supplementary Figures. We added acknowledgments to Francesco Lescai (University of Pavia) for his suggestions on some bioinformatic issues during the revision of the manuscript.

Listed below are our answers to all specific points.

Reviewers' comments:

Reviewer #1 (Remarks to the Author):

THE LOCALIZATION OF CENTROMERE PROTEIN A IS CONSERVED AMONG TISSUES

Cappelletti et al.

Centromere identity is defined epigenetically by the presence of CENP-A, but the highly repetitive satellite DNA occurring at most vertebrate centromeres has--until recently--precluded determination of precise CENP-A position within centromeric chromatin. Circumventing that issue, the authors utilize a previously characterized evolutionarily new horse centromere that lacks alpha satellite or other repetitive DNA to precisely examine CENP-A position in related horses and between tissues of the same horse. Their findings highlight the sliding of CENP-A position at non-repetitive centromere ECA11 between individuals of the same species and the maintenance of those epialleles within tissues of different embryonic origin in the same individual. These findings suggest that individual CENP-A epialleles are determined prior to tissue specification during horse development. While this finding is novel and interesting, the authors do not provide evidence as to when the slide does happen if it doesn't happen during development or in mitotic divisions, but does appear to happen within one generation. The authors suggest that the centromere slide may happen during meiosis, but this suggestion is not followed through. Mechanisms that drive such a slide are not explored or discussed. The genomic re-assembly method of the ECA11 centromere region is concerning, as explained below and can affect the results presented.

Major Points:

COMMENT 1. Given the documented existence of discrete CENP-A binding regions and gaps within the CENP-A binding region at non-repetitive centromeres in humans and horses (Hasson et al., Nat Struct. Mol. Biol. 2013; Nergadze et al., Genome Res. 2018), it is unclear why the authors believed that gaps within CENP-A binding domains found at ECA11 by aligning CENP-A ChIP-seq to EquCab3.0 indicated an issue with the integrity of the EquCab3.0 reference. It's possible that these discrete CENP-A positions represent maternal and paternal CENP-A epialleles at ECA11. For instance, even after reassembly of the ECA11 locus, horses HSF-B and TE still have two distinct CENP-A loci.

ANSWER

We thank the Reviewer for this comment that allowed us to clarify the description of CENP-A binding profiles. In the new version of the manuscript, we are using the term "dips" to define

regions with low coverage of the reads mapped on the reference genome while by “gaps” we mean interruptions (blocks of Ns) in the reference genome.

We observed that, in several horse individuals, the peak profile of the satellite-free centromere of chromosome 11 was irregular (see horse S and horse C in Figure 1) when EquCab3.0 was used as reference. We then carried out a new assembly of this region using our ChIP-seq reads as well as PacBio reads available online. Details on the assembly procedure are now included in the answer to major point 2. The new assembly confirmed that this region was misassembled in EquCab3.0 as described in the answer to major point 2 and in the new version of the manuscript (Results, Lines 113-123: *“In the original EquCab3.0 assembly ... was greatly improved as well as mapping qualities of reads (Supplementary Figure 1).”*). After mapping our ChIP-seq reads to the new assembly, peak shapes were improved and several dips, due to poor quality of the reference, were “curated”. In particular, the small secondary peaks of horse S and C disappeared demonstrating that they did not correspond to epialleles. After mapping the reads on EquCab3.0, horses HSF-B and TE displayed three peaks of different height while, after mapping on EquCab3.0_cen, two peaks of about 100 kb were visible. These two peaks, as also suggested by the Reviewer, likely correspond to epialleles. We better explain this point in the Results section at lines 130-136 (*“Using the EquCab3.0_cen reference, several coverage dips were curated and the peaks become single Gaussian-like peaks covering about 200 kb on the EquCab3.0_cen reference. Horse A shows a broad irregular peak on the EquCab3.0 reference while using the EquCab3.0_cen reference, a more regular and compact peak, occupying about 200 kb, was obtained. After mapping the reads on EquCab3.0, horses HSF-B and TE display three peaks of different height while, after mapping on EquCab3.0_cen, two peaks were observed suggesting that different epialleles for CENP-A binding are present on the two homologous chromosomes.”*).

We would like to underline that the detailed interpretation of the peak profiles does not affect the main conclusion of our work that is the conservation of CENP-A binding domains in different tissues.

COMMENT 2. The authors methodology for improving the sequence of ECA11 in EquCab3.0 is an “iterative chromosome walking approach” that utilizes their CENP-A ChIP-seq reads. However, rebuilding this locus using only reads that bind CENP-A selectively removes any regions/reads that do not bind CENP-A, which could merge discreet CENP-A regions and bias the representation of CENP-A position, and is not necessarily the true genomic assembly of that region. The authors do not provide enough information on how this was performed and therefore it is hard to understand if this is indeed an improvement of the horse ECA11 sequence assembly. It would be helpful to provide more details on this analysis, including a figure depicting what parts of the original assembly are lost in the new EquCab3.0_cen assembly and how CENP-A read positions change between the two assemblies. Specifically, it appears the region between ~27.8 and ~27.9 in the EcuCab3.0 assembly are lost/rearranged in the EcuCab3.0_cen assembly. Are CENP-A reads mapping here lost or relocated to new positions? How does this affect position of CENP-A reads that aligned to this location in the Horse S, D, C, A, HSF-B and TE samples?

Given the plethora of newer sequencing methodologies currently available, a better approach would have been to simply perform a long-read sequencing approach, such as Oxford nanopore sequencing or Pac Bio long-read sequencing to re-assemble that region.

ANSWER

We agree that a description of our chromosome walking approach is needed. We included this information in the Material and Methods section (Lines 302-328: *“The assembly of the CENP-A binding domain from Twilight was performed using an iterative chromosome walking approach ...*

The number of N's per 100 kb in the original sequences and in the newly assembled contig was obtained using Quast Genome assembly Quality (Galaxy Version 5.0.2+galaxy4 or online version available at <http://cab.cc.spbu.ru/quast/>). In the previous version of the manuscript we did not specify that, in addition to ChIP reads, we also used our input reads and the publicly available PacBio (SRR6374292) and Illumina WGS (SRR6374293) reads. We included this information in the new version of the manuscript (Results, Lines 106-112: *"With the goal of determining more precisely the sequence of the centromeric region, we used our paired-end ChIP and input reads and the publicly available PacBio (SRR6374292) and Illumina WGS (SRR6374293) reads to assemble the 617 kb genomic segment containing the CENP-A binding domain of Twilight (NCBI Accession number OQ679756) using an iterative chromosome walking approach, as previously described^{7,13}. We then corrected the EquCab3.0 reference by removing the centromeric locus (chr11:27592872-28352430) and replacing it with the newly assembled sequence."*). We also added a Supplementary Figure (Supplementary Figure 1) reporting the improved mapping quality statistics of both Illumina short reads and PacBio long reads in the centromeric region that we assembled (EquCab3.0_cen) compared to the original EquCab3.0 reference. We added some information on the new assembly. In particular, we curated two sequence gaps whose position and extension (chr11:27808813-27809813 and chr11:28295240-28296240 in EquCab3.0) is specified in the text (Lines 113-123, see answer to comment 1). In addition, we specify that the first gap falls within a coverage dip (Lines 116-118, *"In addition, two sequence gaps (chr11:27808813-27809813 and chr11:28295240-28296240) were present and the first one falls within a coverage dip of the CENP-A peak."*). In the original EquCab3.0 assembly, the region chr11:27707536-27808813 shared high sequence identity with the region chr11:27809814-27911066 and with the entire sequence of the unplaced contig NW_019645621.1. The PacBio long reads and Illumina WGS reads allowed us to demonstrate that at this locus no sequence duplication is present. 8806 reads out of the 16491 reads mapping at chr11:27809814-27911066 in EquCab3.0 were relocated at chr11:27707534-27808741 in EquCab3.0_cen, while the remaining reads were relocated in the unplaced contig. Therefore, as shown in the bottom panel of Figure 1a, in the refined region, the coverage is increased compared to the original reference. Since improvement of the entire horse reference genome was not the goal of our work, we did not modify the unplaced contig. The peak profile of all other horses, was based on the new assembly. We would like to underline that, since the main goal of our work was to test whether peak profiles were maintained in different tissues, our conclusions are not affected by possible minor mis-assemblies of the reference genome.

COMMENT 3. The authors chose to perform cross-linked ChIP-seq for CENP-A instead of native ChIP-sequencing. Since CENP-A is a histone, native ChIP-seq can be done for CENP-A and is the preferred choice that allows titrated MNase digestion, obtaining mono-nucleosome pool (instead of longer stretches of chromatin) before IP, and therefore higher resolution for mapping of CENP-A binding. By choosing fixed chromatin, shearing and mapping of longer chromatin fragments the authors reduce the resolution of CENP-A mapping which is important for their study.

ANSWER

The native ChIP experiments suggested by the Reviewer are not feasible. We obtained cross-linked chromatin samples through the FAANG consortium and it is not possible to obtain new chromatin samples. In addition, in the FAANG consortium, the protocols were standardized. The community decided to use cross-linked chromatin for all ChIP-seq experiments to map transcription factors, histone marks and CENP-A.

We would like to underline that CENP-A nucleosomes are interspersed with canonical H3 nucleosomes and, in our system, CENP-A binding domains span about 100 kb. As described by

Bodor and colleagues (Bodor et al. Elife 2014), a typical human cell has ~400 CENP-A molecules/centromere (200 nucleosomes). From a very careful quantification on a neocentromere where the DNA binding region was known, they showed that less than 10% of centromeric nucleosomes contain CENP-A, the rest is H3. However, they demonstrated that cells do not have identical CENP-A nucleosome distributions, rather different cells have different CENP-A positioning. A similar result was found in the yeast *S. pombe* (Yao et al. JBC 2013) where the authors demonstrated that a relatively small number of CENP-A/Cnp1 nucleosomes are found within the centromeric core and that their positioning relative to underlying DNA varies among genetically homogenous cells. Similar results in yeast and humans indicate that this is a conserved aspect of CENP-A distribution at functional centromeres. Our ChIP experiments were performed on samples of millions of cells. Thus, the experimentally observed CENP-A distribution is a statistical average of CENP-A nucleosome positions in the cell population. In our experimental conditions, performing ChIP-seq using native chromatin will not allow us to increase the resolution of CENP-A mapping.

Finally, as described in our previous work (Nergadze et al. Genome Res 2018), we observed instances of substantial centromere movement, on the order of 50–80 kb, that occurred in a single generation. This is the type of shift that we were expecting to observe if centromere sliding occurred during development. To clarify this point, we added these last sentences to the Discussion (Lines 220-223: *“In addition, we observed instances of substantial centromere movement, on the order of 50–80 kb, that occurred in a single generation. This is the type of shift that we were expecting to observe if centromere sliding occurred during development.”*).

COMMENT 4. The author suggests that the discrete regions of CENP-A binding in some of the horses (Fig. 1B, HSF-B and TE horses) can indicate maternal and paternal alleles. If they have analyzed and reassembled that region of the horse genome using long-read sequencing approaches, that could have led potentially to the identification of possible SNPs in that region, that could have been used to phase the CENP-A ChIP-seq reads and obtain separate maternal or paternal CENP-A binding patterns.

ANSWER

As underlined by the Reviewer, in our previous work, the presence of epialleles was demonstrated using a SNP approach (Purgato et al. Chromosoma 2015, Nergadze et al. Genome Res. 2018). In the present work, we were unable to use the same approach for horses HSF-B and TE because of the low coverage of our input datasets. However, for horse HFS-B, epialleles were previously detected using a SNP approach following ChIP-on-chip experiments (Purgato et al. Chromosoma 2015). Since the main goal of our work was to test whether peak profiles were maintained in different tissues, our conclusions are not affected by the position of CENP-A binding domains and by the precise identification and characterization of epialleles. Therefore, these aspects are not discussed in the manuscript.

COMMENT 5. The emergence of evolutionarily new centromeres like ECA11 is thought to occur at regions where DNA is fragile or frequently transcribed, allowing opportunities for incorporation of CENP-A. Though ECA11 is not a repetitive centromere, other evolutionarily young horse centromeres that do not occur at satellite DNA have acquired repetitive elements and previous work suggests that ECA11 is likely to also gain repetitive DNA elements as it continues to mature. This, and the authors finding that the AH1 mare may carry a chromosomal rearrangement at the ECA11 locus suggests that this region may also be fragile or prone to DNA rearrangements. Given

this knowledge, the authors should discuss the potential differences in the underlying DNA sequences between different horses at the ECA11 locus and how this may bias alignments of CENP-A ChIP-seq data from other horses to the Twilight-derived EcuCab3.0 or EcuCab3.0_cen assemblies.

ANSWER

We thank the Reviewer for the insights into the possible role of DNA fragility in the rearrangement of the ECA11 centromeric region. It was proposed that the emergence of new centromeres during evolution may be triggered by DNA fragility. Since sites of breakage are recurrent during evolution and some of them tend to be used for centromere formation (Murphy et al. Science 2005, Longo et al. BMC Genomics 2009), it is possible that, also at ECA11, DNA breaks and rearrangements may have occurred. If this was the case, we may expect some variability, due to sequence rearrangement of this region, in the horse population, which may bias alignments of CENP-A ChIP-seq data from other horses to the Twilight-derived EquCab3.0 or EquCab3.0_cen assemblies. These considerations were included in the Discussion (Lines 208-216: *“Similarly, we cannot exclude that some of the peak irregularities of the other horses may be due to specific sequence rearrangements compared to the reference genome.*

It was proposed that the emergence of new centromeres during evolution may be triggered by DNA fragility. Since sites of breakage are recurrent during evolution and some of them tend to be used for centromere formation^{34,35}, it is possible that, also at ECA11, DNA breaks and rearrangements may have occurred. If this was the case, we may expect some variability, due to sequence rearrangement of this region, in the horse population, which may bias alignments of CENP-A ChIP-seq data from other horses to the Twilight-derived EquCab3.0 or EquCab3.0_cen assemblies.”)

We previously suggested that during evolution satellite-free centromere may start the maturation process through the acquisition of duplication (Nergadze et al. Genome Res. 2018) followed by satellite DNA formation. It is possible that chromosomal rearrangements may represent the first step of this maturation process towards the acquisition of satellite DNA.

COMMENT 6. In figure 2, there seems to be some small-scale differences or sliding of CENP-A position between fibroblasts and testis samples of stallions AH3 and AH4. It would be helpful if the authors provided a track of significantly enriched CENP-A peaks relative to background to help clarify this. Differences in CENP-A position between fibroblasts and testis (does this sample contain spermatozoa?) might be important for helping to define when CENP-A position changes between horse generations.

ANSWER

The observation of the Reviewer is correct. To prepare chromatin from fibroblasts we used 50 millions of cultured cells while the amount of chromatin that we could obtain from the tissues was relatively low. This is the reason why, as also noticed by Reviewer 2, in tissue samples the background is generally higher than in fibroblasts. A sentence on this point was added to the Materials and Methods section (Lines 275-279: *“Chromatin from about 50 million primary fibroblasts was cross-linked with 1% formaldehyde, extracted, and sonicated to obtain DNA fragments ranging from 200 to 800 bp. Chromatin from brain, lamina, liver, ovary and testis was extracted and sheared by Diagenode ChIP-Seq Profiling Service (Diagenode, Cat# G02010000, Liège, Belgium) as previously described^{26,28}. For ChIP experiments we used variable amount of chromatin ranging from 4 to 13 µg.”*).

We added a Figure (Supplementary Figure 2) where all peak profiles are shown on the same scale and peak calling tracks are reported. As also shown by the peak calling tracks, the CENP-A peak from the testis of stallion AH3 shows a tail which is not detected in the other tissues. We cannot definitely conclude whether this tail is part of the background or is due to the presence of a sub-population of cells in which minor sliding may have occurred. It is tempting to speculate that this sliding, if any, may be due to the presence of mature spermatozoa and cells at different stages of meiosis that are present in the testis samples, supporting the hypothesis that sliding occurs in meiosis. This sentence was added to the Discussion (Lines 248-253: *“It is worth noticing that the CENP-A peak from the testis of stallion AH3 shows a tail which is not detected in the other tissues. We cannot definitely conclude whether this tail is part of the background or is due to the presence of a sub-population of cells in which sliding may have occurred. It is tempting to speculate that the shape of the CENP-A binding domain from the testis of AH3 may be due to the presence of a fraction of mature spermatozoa and cells at different stages of meiosis in which sliding may have occurred.”*).

COMMENT 7. The authors found that there is no difference in CENP-A binding in different tissues of the same individual suggesting that CENP-A binding pattern is determined prior to tissue differentiation. This is an interesting and novel finding. However, the authors showed that centromere sliding can happen in one generation (compare CENP-A binding pattern of Twilight’s fibroblasts in Fig. 1a with the pattern of her son AH4 in Fig. 2a), and previously showed that the centromere position is stable during mitotic divisions, suggesting that the sliding may occur during meiosis. The authors did not explore further this option.

ANSWER

An interesting development of the work presented here will be to compare CENP-A binding profiles from samples of spermatozoa with other tissues. However, a clear answer on the possibility of sliding in meiosis could come from ChIP-seq on single spermatozoa and oocytes which will be the aim of future work as these were not collected from the horses used in this study.

Minor Points:

COMMENT

1. The authors state in lines 57-59 that ECA11 is the first centromere devoid of alpha satellite DNA to be detected in vertebrates. However, many human centromeres devoid of alpha satellite DNA have been documented prior to this finding (Amor and Choo, *Am J Hum Genet.* 2002), with the first being in 1993 (Voullaire et al., *Am J Hum Genet.*).

ANSWER

The ECA11 centromere was the first satellite-free centromere, stably present in a vertebrate species as a normal component of its karyotype, to be discovered. We clarified this point in the Introduction. We also added a sentence on human neocentromeres with reference to relevant literature (Lines 57-60: *“Satellite-free neocentromeres have been previously described in sporadic human clinical samples^{2,3,21,22} while the ECA11 centromere was the first centromere devoid of satellite DNA to be found stably present in a vertebrate species, demonstrating that a natural centromere can exist without satellite DNA¹¹.”*).

COMMENT

2. It would have been nice to show the CENP-A binding pattern of Twilight's fibroblasts (Fig. 1a) directly above the pattern of her son (Fig. 2a, b) to better present the centromere sliding within one generation.

ANSWER:

In the absence of the pattern of the father of horse AH4, we cannot draw any conclusion on epiallele inheritance or sliding between Twilight and her son.

Reviewer #2 (Remarks to the Author):

COMMENT

This manuscript investigates centromere positioning and inheritance in horses. Uniquely, horses harbour a satellite free centromere on chromosome 11, mapped by the binding of the centromeric histone CENP-A. The discovery of this natural satellite free centromere has enabled investigations into centromere specification and transgenerational inheritance at the genomic level. It was previously shown that the domain of CENP-A binding is inherited, but that it can slide within a 500 kb window in a single generation. As centromere sliding is not observed in mitotic cells in culture, the authors propose that it occurs in meiosis or embryogenesis. To address this possibility, the authors investigate whether CENP-A position changes in differentiated tissues of the same animal. Overall, the data supports the conclusion that CENP-A position does not change, however a more detailed description of CENP-A profile in the investigated tissues would be informative here (see point 2 below).

This study is well written and presented and the experiments are carried out at a high standard. It presents two major findings:

First, presented in Figure 1, the authors use newly generated CENP-A paired end CHIP seq data to redefine the centromeric region of chromosome 11 in the latest release of the Twilight horse reference genome (EquCab3.0). Based on maps generated for Twilight and 6 additional horses, this new and corrected centromere 11 reference sequence, called EquCab3.0_cen, contains more regular CENP-A peaks that are less interrupted and more Gaussian in shape. Going forward, this is an important resource for the community.

Second, to test whether centromere position can slide during tissue differentiation, the authors map and compare CENP-A position by CHIP-seq in five tissues from four individual horses (extensively phenotyped as part of the FAANG project). The tissue selected were from different embryonic origin – endodermal (liver), mesodermal (ovary/testes) and ectodermal (lamina and brain). Results presented in Figure 2b showed that, in general, the position of the centromere did not move compared to the control fibroblast cell line. However, there does appear to be more 'background signal' in all data from tissues, particularly in the case of the brain and ovaries/tissues. Can the authors comment on this? Also in the ovary and tissue samples, is it possible that these tissues contain meiotic cells? This would be important to know, especially as the CENP-A signal appears to occupy a larger domain with less sharp peaks e.g. sample AH4.

ANSWER

We thank the Reviewer for the appreciation of our work and for the insights into meiotic cells. The questions raised by this Reviewer are in line with major point 6 and 7 from Reviewer 1. As stated in our answer to Reviewer 1, to prepare chromatin from fibroblasts we used 50 million cultured cells while the amount of chromatin that we could obtain from the tissues was relatively low. This is the reason why, in tissue samples, the background is generally higher than in fibroblasts. This is particularly evident in the brain from all horses and in the testis from both stallions. A sentence on this point was added to the Material and Methods section (Lines 275-279: *“Chromatin from about 50 million primary fibroblasts was cross-linked with 1% formaldehyde, extracted, and sonicated to obtain DNA fragments ranging from 200 to 800 bp. Chromatin from brain, lamina, liver, ovary and testis was extracted and sheared by Diagenode ChIP-Seq Profiling Service (Diagenode, Cat# G02010000, Liège, Belgium) as previously described^{26,28}. For ChIP experiments we used variable amount of chromatin ranging from 4 to 13 µg.”*).

Both in the testis and in the ovary tissue, meiotic cells were present. It is tempting to speculate that the shape of the CENP-A binding domain from the testis of AH3 may be due to the presence of a fraction of mature spermatozoa and cells at different stages of meiosis in which sliding may have occurred. This sentence was added to the Discussion (Lines 248-253: *“It is worth noticing that the CENP-A peak from the testis of stallion AH3 shows a tail which is not detected in the other tissues. We cannot definitely conclude whether this tail is part of the background or is due to the presence of a sub-population of cells in which sliding may have occurred. It is tempting to speculate that the shape of the CENP-A binding domain from the testis of AH3 may be due to the presence of a fraction of mature spermatozoa and cells at different stages of meiosis in which sliding may have occurred.”*).

REVIEWERS' COMMENTS:

Reviewer #1 (Remarks to the Author attached)

Reviewer #2 (Remarks to the Author):

In this improved version of the manuscript, the authors have addressed my previous comments with additions to the text, clarifications to the materials and methods and the added supplementary figure.

We thank Reviewer 1 for the careful analysis of the revised manuscript that allowed us to further improve it. The corrections to the revised version of the manuscript suggested by Reviewer 1 refer to points 6 and 7 and our answer to the new comments of the Reviewer are reported here below in blue.

The new modifications of the manuscript are reported in red in the second revised version of the manuscript.

As suggested by the Reviewer, we moved Supplementary Figure 2b to the main text as Figure 2c.

Reviewers' comments:

Reviewer #1 (Remarks to the Author):

THE LOCALIZATION OF CENTROMERE PROTEIN A IS CONSERVED AMONG TISSUES

Cappelletti et al.

Centromere identity is defined epigenetically by the presence of CENP-A, but the highly repetitive satellite DNA occurring at most vertebrate centromeres has--until recently--precluded determination of precise CENP-A position within centromeric chromatin. Circumventing that issue, the authors utilize a previously characterized evolutionarily new horse centromere that lacks alpha satellite or other repetitive DNA to precisely examine CENP-A position in related horses and between tissues of the same horse. Their findings highlight the sliding of CENP-A position at non- repetitive centromere ECA11 between individuals of the same species and the maintenance of those epialleles within tissues of different embryonic origin in the same individual. These findings suggest that individual CENP-A epialleles are determined prior to tissue specification during horse development. While this finding is novel and interesting, the authors do not provide evidence as to when the slide does happen if it doesn't happen during development or in mitotic divisions, but does appear to happen within one generation. The authors suggest that the centromere slide may happen during meiosis, but this suggestion is not followed through. Mechanisms that drive such a slide are not explored or discussed. The genomic reassembly method of the ECA11 centromere region is concerning, as explained below and can affect the results presented.

Major Points:

COMMENT 1. Given the documented existence of discrete CENP-A binding regions and gaps within the CENP-A binding region at non-repetitive centromeres in humans and horses (Hasson et al., Nat Struct. Mol. Biol. 2013; Nergadze et al., Genome Res. 2018), it is unclear why the authors believed that gaps within CENP-A binding domains found at ECA11 by aligning CENP-A ChIP-seq to EquCab3.0 indicated an issue with the integrity of the EquCab3.0 reference. It's possible that these discrete CENP-A positions represent maternal and paternal CENP-A epialleles at ECA11. For instance, even after reassembly of the ECA11 locus, horses HSF-B and TE still have two distinct CENP-A loci.

ANSWER

We thank the Reviewer for this comment that allowed us to clarify the description of CENP-A binding profiles. In the new version of the manuscript, we are using the term "dips" to define regions with low coverage of the reads mapped on the reference genome while by "gaps" we mean interruptions (blocks of Ns) in the reference genome.

We observed that, in several horse individuals, the peak profile of the satellite-free centromere of chromosome 11 was irregular (see horse S and horse C in Figure 1) when EquCab3.0 was used as reference. We then carried out a new assembly of this region using our ChIP-seq reads as well as PacBio reads available online. Details on the assembly procedure are now included in the answer to major point 2. The new assembly confirmed that this region was misassembled in EquCab3.0 as described in the answer to major point 2 and in the new version of the manuscript (Results, Lines 113-123: "In the original EquCab3.0 assembly ... was greatly improved as well as mapping qualities of reads (Supplementary Figure 1)."). After mapping our ChIP-seq reads to the new assembly, peak shapes were improved and several dips, due to poor quality of the reference, were "curated". In particular, the small secondary peaks of horse S and C disappeared demonstrating that they did not correspond to epialleles. After mapping the reads on EquCab3.0, horses HSF-B and TE displayed three peaks of different height while, after mapping on EquCab3.0_cen, two peaks of about 100 kb were visible. These two peaks, as also suggested by the Reviewer, likely correspond to epialleles. We better explain this point in the Results

section at lines 130-136 (“Using the EquCab3.0_cen reference, several coverage dips were curated and the peaks become single Gaussian-like peaks covering about 200 kb on the EquCab3.0_cen reference. Horse A shows a broad irregular peak on the EquCab3.0 reference while using the EquCab3.0_cen reference, a more regular and compact peak, occupying about 200 kb, was obtained. After mapping the reads on EquCab3.0, horses HSF-B and TE display three peaks of different height while, after mapping on EquCab3.0_cen, two peaks were observed suggesting that different epialleles for CENP-A binding are present on the two homologous chromosomes.”).

We would like to underline that the detailed interpretation of the peak profiles does not affect the main conclusion of our work that is the conservation of CENP-A binding domains in different tissues.

Reviewer Response: Authors have addressed this comment well in the revised version. This new information that is provided is very relevant to the correct interpretation of the data. The inclusion of available PacBio reads for this region/assembly adds much more confidence to the authors re-assembly of this region, and the expanded methods and details provide valuable information for the reader. Further, the authors clarification of "gaps" and "dips" provides additional insight into why the authors originally interpreted the assembly to EquCab3.0 to be problematic.

COMMENT 2. The authors methodology for improving the sequence of ECA11 in EquCab3.0 is an “iterative chromosome walking approach” that utilizes their CENP-A ChIP-seq reads. However, rebuilding this locus using only reads that bind CENP-A selectively removes any regions/reads that do not bind CENP-A, which could merge discreet CENP-A regions and bias the representation of CENP-A position, and is not necessarily the true genomic assembly of that region. The authors do not provide enough information on how this was performed and therefore it is hard to understand if this is indeed an improvement of the horse ECA11 sequence assembly. It would be helpful to provide more details on this analysis, including a figure depicting what parts of the original assembly are lost in the new EquCab3.0_cen assembly and how CENP-A read positions change between the two assemblies. Specifically, it appears the region between ~27.8 and ~27.9 in the EcuCab3.0 assembly are lost/rearranged in the EcuCab3.0_cen assembly. Are CENP-A reads mapping here lost or relocated to new positions? How does this affect position of CENP-A reads that aligned to this location in the Horse S, D, C, A, HSF-B and TE samples?

Given the plethora of newer sequencing methodologies currently available, a better approach would have been to simply perform a long-read sequencing approach, such as Oxford nanopore sequencing or Pac Bio long-read sequencing to re-assemble that region.

ANSWER

We agree that a description of our chromosome walking approach is needed. We included this information in the Material and Methods section (Lines 302-328: “The assembly of the CENP-A binding domain from Twilight was performed using an iterative chromosome walking approach ... The number of N’s per 100 kb in the original sequences and in the newly assembled contig was obtained using Quast Genome assembly Quality (Galaxy Version 5.0.2+galaxy4 or online version available at <http://cab.cc.spbu.ru/quast/>).”). In the previous version of the manuscript we did not specify that, in addition to ChIP reads, we also used our input reads and the publicly available PacBio (SRR6374292) and Illumina WGS (SRR6374293) reads. We included this information in the new version of the manuscript (Results, Lines 106-112: “With the goal of determining more precisely the sequence of the centromeric region, we used our paired-end ChIP and input reads and the publicly available PacBio (SRR6374292) and Illumina WGS (SRR6374293)reads to assemble the 617 kb genomic segment containing the CENP-A binding domain of Twilight (NCBI Accession number OQ679756) using an iterative chromosome walking approach, as previously described 7,13. We then corrected the EquCab3.0 reference by removing the centromeric locus (chr11:27592872- 28352430) and replacing it with the newly assembled sequence.”). We also added a Supplementary Figure (Supplementary Figure 1) reporting the improved mapping quality statistics of both Illumina short reads and PacBio long reads in the centromeric region that we assembled (EquCab3.0_cen) compared to the original EquCab3.0 reference. We added some information on the new assembly. In particular, we curated two sequence gaps whose position and extension (chr11:27808813-27809813 and chr11:28295240-28296240 in EquCab3.0) is specified in the text (Lines

113-123, see answer to comment 1). In addition, we specify that the first gap falls within a coverage dip (Lines 116-118, “In addition, two sequence gaps (chr11:27808813-27809813 and chr11:28295240-28296240) were present and the first one falls within a coverage dip of the CENP- A peak.”). In the original EquCab3.0 assembly, the region chr11:27707536-27808813 shared high sequence identity with the region chr11:27809814-27911066 and with the entire sequence of the unplaced contig NW_019645621.1. The PacBio long reads and Illumina WGS reads allowed us to demonstrate that at this locus no sequence duplication is present. 8806 reads out of the 16491 reads mapping at chr11:27809814-27911066 in EquCab3.0 were relocated at chr11:27707534- 27808741 in EquCab3.0_cen, while the remaining reads were relocated in the unplaced contig. Therefore, as shown in the bottom panel of Figure 1a, in the refined region, the coverage is increased compared to the original reference. Since improvement of the entire horse reference genome was not the goal of our work, we did not modify the unplaced contig. The peak profile of all other horses, was based on the new assembly. We would like to underline that, since the main goal of our work was to test whether peak profiles were maintained in different tissues, our conclusions are not affected by possible minor mis-assemblies of the reference genome.

Reviewer response: As in the previous comment, these details are very important for the manuscript and help clarify the author’s approach for resolving an assembly issue in EquCab3.0. This is a major improvement on the methods and reasoning detailed in the original manuscript.

COMMENT 3. The authors chose to perform cross-linked ChIP-seq for CENP-A instead of native ChIPsequencing.

Since CENP-A is a histone, native ChIP-seq can be done for CENP-A and is the preferred choice that allows titrated MNase digestion, obtaining mono-nucleosome pool (instead of longer stretches of chromatin) before IP, and therefore higher resolution for mapping of CENP- A binding. By choosing fixed chromatin, shearing and mapping of longer chromatin fragments the authors reduce the resolution of CENP-A mapping which is important for their study.

ANSWER

The native ChIP experiments suggested by the Reviewer are not feasible. We obtained cross-linked chromatin samples through the FAANG consortium and it is not possible to obtain new chromatin samples. In addition, in the FAANG consortium, the protocols were standardized. The community decided to use cross-linked chromatin for all ChIP-seq experiments to map transcription factors, histone marks and CENP-A.

We would like to underline that CENP-A nucleosomes are interspersed with canonical H3 nucleosomes and, in our system, CENP-A binding domains span about 100 kb. As described by Bodor and colleagues (Bodor et al. *Elife* 2014), a typical human cell has ~400 CENP-A molecules/centromere (200 nucleosomes). From a very careful quantification on a neocentromere where the DNA binding region was known, they showed that less than 10% of centromeric nucleosomes contain CENP-A, the rest is H3. However, they demonstrated that cells do not have identical CENP-A nucleosome distributions, rather different cells have different CENP-A positioning. A similar result was found in the yeast *S. pombe* (Yao et al. *JBC* 2013) where the authors demonstrated that a relatively small number of CENP-A/Cnp1 nucleosomes are found within the centromeric core and that their positioning relative to underlying DNA varies among genetically homogenous cells. Similar results in yeast and humans indicate that this is a conserved aspect of CENP-A distribution at functional centromeres. Our ChIP experiments were performed on samples of millions of cells. Thus, the experimentally observed CENP-A distribution is a statistical average of CENP-A nucleosome positions in the cell population. In our experimental conditions, performing ChIP-seq using native chromatin will not allow us to increase the resolution of CENP-A mapping.

Finally, as described in our previous work (Nergadze et al. *Genome Res* 2018), we observed instances of substantial centromere movement, on the order of 50–80 kb, that occurred in a single generation. This is the type of shift that we were expecting to observe if centromere sliding occurred during development. To clarify this point, we added these last sentences to the Discussion (Lines 220-223: “In addition, we observed instances of substantial centromere movement, on the order of 50–80 kb, that occurred in a

single generation. This is the type of shift that we were expecting to observe if centromere sliding occurred during development.”).

Reviewer response: it is reasonable to be limited by the type of data/samples (in this case, crosslinked chromatin fragments) that are available for particular horses. The authors point that resolving CENP-A position is limited when taking a population-based approach, like CHIP-seq, is valid.

COMMENT 4. The author suggests that the discrete regions of CENP-A binding in some of the horses (Fig. 1B, HSF-B and TE horses) can indicate maternal and paternal alleles. If they have analyzed and reassembled that region of the horse genome using long-read sequencing approaches, that could have led potentially to the identification of possible SNPs in that region, that could have been used to phase the CENP-A ChIP-seq reads and obtain separate maternal or paternal CENP-A binding patterns.

ANSWER

As underlined by the Reviewer, in our previous work, the presence of epialleles was demonstrated using a SNP approach (Purgato et al. Chromosoma 2015, Nergadze et al. Genome Res. 2018). In the present work, we were unable to use the same approach for horses HSF-B and TE because of the low coverage of our input datasets. However, for horse HFS-B, epialleles were previously detected using a SNP approach following ChIP-on-chip experiments (Purgato et al. Chromosoma 2015). Since the main goal of our work was to test whether peak profiles were maintained in different tissues, our conclusions are not affected by the position of CENP-A binding domains and by the precise identification and characterization of epialleles. Therefore, these aspects are not discussed in the manuscript.

Reviewer response: I accept the reasoning of the authors. Since this is not the primary concern of this manuscript, and because the authors also address the possible presence of epialleles in their response to Comment 1, this is sufficiently addressed.

COMMENT 5. The emergence of evolutionarily new centromeres like ECA11 is thought to occur at regions where DNA is fragile or frequently transcribed, allowing opportunities for incorporation of CENPA. Though ECA11 is not a repetitive centromere, other evolutionarily young horse centromeres that do not occur at satellite DNA have acquired repetitive elements and previous work suggests that ECA11 is likely to also gain repetitive DNA elements as it continues to mature. This, and the authors finding that the AH1 mare may carry a chromosomal rearrangement at the ECA11 locus suggests that this region may also be fragile or prone to DNA rearrangements. Given this knowledge, the authors should discuss the potential differences in the underlying DNA sequences between different horses at the ECA11 locus and how this may bias alignments of CENP-A ChIP-seq data from other horses to the Twilight-derived EcuCab3.0 or EcuCab3.0_cen assemblies.

ANSWER

We thank the Reviewer for the insights into the possible role of DNA fragility in the rearrangement of the ECA11 centromeric region. It was proposed that the emergence of new centromeres during evolution may be triggered by DNA fragility. Since sites of breakage are recurrent during evolution and some of them tend to be used for centromere formation (Murphy et al. Science 2005, Longo et al. BMC Genomics 2009), it is possible that, also at ECA11, DNA breaks and rearrangements may have occurred. If this was the case, we may expect some variability, due to sequence rearrangement of this region, in the horse population, which may bias alignments of CENP-A ChIP-seq data from other horses to the Twilight-derived EcuCab3.0 or EcuCab3.0_cen assemblies.

These considerations were included in the Discussion (Lines 208-216: “Similarly, we cannot exclude that some of the peak irregularities of the other horses may be due to specific sequence rearrangements compared to the reference genome. It was proposed that the emergence of new centromeres during evolution may be triggered by DNA fragility. Since sites of breakage are recurrent during evolution and some of them tend to be used for centromere formation 34,35, it is possible that, also at ECA11, DNA breaks and rearrangements may have occurred. If this was the case, we may expect some variability, due to sequence rearrangement of this region, in the horse population, which may bias alignments of CENP-A ChIP-seq data from other horses to the Twilight-derived EcuCab3.0 or EcuCab3.0_cen assemblies.”)

We previously suggested that during evolution satellite-free centromere may start the maturation

process through the acquisition of duplication (Nergadze et al. Genome Res. 2018) followed by satellite DNA formation. It is possible that chromosomal rearrangements may represent the first step of this maturation process towards the acquisition of satellite DNA.

Reviewer response: The authors addressed my concerns well regarding the possible role of fragility at the ECA11 centromere.

COMMENT 6. In figure 2, there seems to be some small-scale differences or sliding of CENP-A position between fibroblasts and testis samples of stallions AH3 and AH4. It would be helpful if the authors provided a track of significantly enriched CENP-A peaks relative to background to help clarify this. Differences in CENP-A position between fibroblasts and testis (does this sample contain spermatozoa?) might be important for helping to define when CENP-A position changes between horse generations.

ANSWER

The observation of the Reviewer is correct. To prepare chromatin from fibroblasts we used 50 millions of cultured cells while the amount of chromatin that we could obtain from the tissues was relatively low. This is the reason why, as also noticed by Reviewer 2, in tissue samples the background is generally higher than in fibroblasts. A sentence on this point was added to the Materials and Methods section (Lines 275-279: "Chromatin from about 50 million primary fibroblasts was cross-linked with 1% formaldehyde, extracted, and sonicated to obtain DNA fragments ranging from 200 to 800 bp. Chromatin from brain, lamina, liver, ovary and testis was extracted and sheared by Diagenode ChIP-Seq Profiling Service (Diagenode, Cat# G02010000, Liège, Belgium) as previously described 26,28. For ChIP experiments we used variable amount of chromatin ranging from 4 to 13 g.").

We added a Figure (Supplementary Figure 2) where all peak profiles are shown on the same scale and peak calling tracks are reported. As also shown by the peak calling tracks, the CENP-A peak from the testis of stallion AH3 shows a tail which is not detected in the other tissues. We cannot definitely conclude whether this tail is part of the background or is due to the presence of a sub-population of cells in which minor sliding may have occurred. It is tempting to speculate that this sliding, if any, may be due to the presence of mature spermatozoa and cells at different stages of meiosis that are present in the testis samples, supporting the hypothesis that sliding occurs in meiosis. This sentence was added to the Discussion (Lines 248-253: "It is worth noticing that the CENP-A peak from the testis of stallion AH3 shows a tail which is not detected in the other tissues. We cannot definitely conclude whether this tail is part of the background or is due to the presence of a sub-population of cells in which sliding may have occurred. It is tempting to speculate that the shape of the CENP-A binding domain from the testis of AH3 may be due to the presence of a fraction of mature spermatozoa and cells at different stages of meiosis in which sliding may have occurred.").

Reviewer response: The authors insistence that the goal of this manuscript is to address whether peak profiles of CENP-A are different in various tissues is at odds with the authors lack of rigorous analysis of CENP-A position between tissues of the same horse. It is not ideal to compare coverage between samples prepared from varying input levels, as it can manifest into issues during peak calling. Peaks should be called from scaled samples to help mitigate variable background peak detection in samples with different levels of coverage. Supplementary Figure 2 is very important to understanding Figure 2 and should be incorporated as part of the main Figure 2. And I wonder why this was not available already in the original version.

Regardless, the peak patterns observed in ovary/testis of horses AH3 and AH4 are noticeably different than patterns observed in other tissues of these same horses. Importantly, it lowers the authors claims because it raises concerns whether the binding pattern is indeed changing between tissues, as we now see that the pattern is different between testis and other tissues. so, the main question is clearly not answered. These differences and changes should be discussed in more detail if the goal of this paper is to evaluate changes in CENP-A position among different tissues of the same horse. Moreover, comparing CENP-A position among the tissues of single horses may not be sufficient to explain how CENP-A position may slide from one generation to the next, which leaves the underlying question of "how does CENP-A position drastically slide between horses in a single generation?" unanswered.

ANSWER: it seems that, by “scaled samples”, the Reviewer is referring to the different scales on the y axis in Figure 2. The scale of the y axis is not the same to highlight the position of the peak rather than its height. Accordingly, the scale of the x axis is the same across all tissues of the same horse. We added a sentence in the figure legend to clarify this point (Lines 560-561: “The scale of the y axis is not the same across samples to highlight the position of the peak rather than its height.”).

To further address the concerns of the Reviewer, it is also important to underline that the enrichment peaks shown in Figure 2b were obtained using bamCompare performing a RPKM normalization in subtractive mode on the input sample, as described in the Methods section. Furthermore, the algorithm of the peak calling tool (SICER2) takes into account the number of reads of the different samples. In the new version of the manuscript, we better explained peak calling parameters in the Methods section (Lines 305-306: “Peak calling was performed using SICER2⁵⁴ using -w 200 and -g 1000 parameters and filtering for islands with FDR less than or equal to 0.01.”). As suggested by the Reviewer, we included the peak calling tracks (previously shown in Supplementary Figure 2b) in Figure 2 (new panel c). We maintained the figure showing all peak profiles on the same scale (previous Supplementary Figure 2a) in the Supplementary Material (new Supplementary Figure 2). This figure was not present in the original version of the manuscript because we did not realize that it would help clarifying our findings.

The tail of the CENP-A peak profile in the testis sample from stallion AH3 has been already addressed in the Discussion. The fact that a slight difference in the shape of the CENP-A peak is restricted to stallion AH3 in a tissue containing meiotic cells supports the conclusion that centromere sliding does occur in meiosis, as already discussed in the first revision. However, the peak position of the testis from AH4 is not noticeably different from the other tissues of this individual.

The Reviewer observed that comparing CENP-A position among the tissues of single horses may not be sufficient to explain how CENP-A position may slide from one generation to the next. This is true but the goal of the manuscript is not to understand how CENP-A position can slide in a single generation. As already mentioned in the Introduction and in the Discussion, centromere sliding was already investigated by us using hybrid families of horses and donkeys (Nergadze et al. Genome Research 2018).

The goal of the present manuscript was to test whether centromere sliding occurs during development. The conclusion of the work is that centromere position is conserved in different tissues from the same individual, suggesting that the phenomenon of centromere sliding does not occur during tissues differentiation but rather during meiosis. A clear answer on the possibility of sliding in meiosis could come from CHIP-seq on single spermatozoa and oocytes which will be the aim of future work as these were not collected from the horses used in this study. As suggested by the Reviewer, we added a sentence in the Discussion section to clarify this point and explain which are the future experiments which can be planned to better address this issue (Lines 254-258: “An interesting development of the work presented here will be to compare CENP-A binding profiles from samples of spermatozoa with other tissues. However, a clear answer on the possibility of sliding in meiosis could come from CHIP-seq on single spermatozoa and oocytes which will be the aim of future work.”).

COMMENT 7. The authors found that there is no difference in CENP-A binding in different tissues of the same individual suggesting that CENP-A binding pattern is determined prior to tissue differentiation. This is an interesting and novel finding. However, the authors showed that centromere sliding can happen in one generation (compare CENP-A binding pattern of Twilight’s fibroblasts in Fig. 1a with the pattern of her son AH4 in Fig. 2a), and previously showed that the centromere position is stable during mitotic divisions, suggesting that the sliding may occur during meiosis. The authors did not explore further this option.

ANSWER

An interesting development of the work presented here will be to compare CENP-A binding profiles from samples of spermatozoa with other tissues. However, a clear answer on the possibility of sliding in meiosis could come from CHIP-seq on single spermatozoa and oocytes which will be the aim of future work as these were not collected from the horses used in this study.

Reviewer response: I understand that spermatozoa and oocytes were not collected. However, the main question remains unanswered, reducing the impact and novelty of the results. I still think the manuscript is not complete as is. Although not ideal, inclusion of these proposed experiments in the discussion of the paper would help to better address the next steps needed to clearly define how CENP-A positional sliding occurs in a single generation and would increase the potential impact of the paper.

ANSWER: as mentioned in the previous answer, we added a sentence in the Discussion section to clarify this point and explain which are the future experiments to better address this issue (Lines 254-258).

Minor Points:

COMMENT

1. The authors state in lines 57-59 that ECA11 is the first centromere devoid of alpha satellite DNA to be detected in vertebrates. However, many human centromeres devoid of alpha satellite DNA have been documented prior to this finding (Amor and Choo, Am J Hum Genet. 2002), with the first being in 1993 (Voullaire et al., Am J Hum Genet.).

ANSWER

The ECA11 centromere was the first satellite-free centromere, stably present in a vertebrate species as a normal component of its karyotype, to be discovered. We clarified this point in the Introduction. We also added a sentence on human neocentromeres with reference to relevant literature (Lines 57-60: "Satellite-free neocentromeres have been previously described in sporadic human clinical samples 2,3,21,22 while the ECA11 centromere was the first centromere devoid of satellite DNA to be found stably present in a vertebrate species, demonstrating that a natural centromere can exist without satellite DNA 11.").

Reviewer response: This is suitably clarified in the introduction.

COMMENT

2. It would have been nice to show the CENP-A binding pattern of Twilight's fibroblasts (Fig. 1a) directly above the pattern of her son (Fig. 2a, b) to better present the centromere sliding within one generation.

ANSWER:

In the absence of the pattern of the father of horse AH4, we cannot draw any conclusion on epiallele inheritance or sliding between Twilight and her son.

Reviewer response: This is a safe conclusion and it's unfortunate that we do not have additional data from additional experiments to help inform on centromere sliding within one generation.